# The Immunogenicity of Glutaraldehyde Inactivated PTx Is Determined by the Quantity of Neutralizing Epitopes

**DOI:** 10.3390/vaccines13080817

**Published:** 2025-07-31

**Authors:** Xi Wang, Xinyue Cui, Chongyang Wu, Ke Tao, Shuyuan Pan, Wenming Wei

**Affiliations:** 1Lab of Novel Bacterial Vaccines Research and Development, Department of Research and Development, Beijing Institute of Biological Products Co., Ltd., Beijing 100176, China; wangqian165@sinopharm.com (X.W.); cuixinyue3@sinopharm.com (X.C.); wuchongyang@sinopharm.com (C.W.); 2Engineering Research Center of Antibodies, Department of Research and Development, Beijing Institute of Biological Products Co., Ltd., Beijing 100176, China; taoke@sinopharm.com; 3Project Management Office, Department of Research and Development, Beijing Institute of Biological Products Co., Ltd., Beijing 100176, China

**Keywords:** pertussis toxin, glutaraldehyde detoxification, acellular pertussis vaccines, antigenic properties, immunogenicity

## Abstract

Background/Objectives: Chemically or genetically detoxified pertussis toxin (PTx) is a crucial antigen component of the acellular pertussis vaccine. Chemical detoxification using glutaraldehyde generally causes significant structural changes to the toxin. However, how these structural changes in PTx affect its antigenic properties remains unclear. Additionally, there is limited knowledge regarding how many alterations in antigenic properties impact immunogenicity. Methods: To investigate the impact of structural changes on antigenic properties, we developed a sandwich ELISA to quantify the neutralizing epitopes on PTx. Subsequently, we analyzed different PTx toxoid (PTd) preparations with the assay. Additionally, we assessed the immunogenicity of various acellular pertussis vaccine candidates containing these PTd preparations. Finally, the assay was applied to evaluate the consistency of commercial batches of PTx and PTd intermediates. Results: The assay demonstrated reasonable specificity, accuracy, and precision, and it was sensitive enough to quantify variations in neutralizing epitopes among different PTd samples that shared the same protein concentration. Importantly, we found a positive correlation between the number of neutralizing epitopes in detoxified PTx and its immunogenicity, indicating that the amount of neutralizing epitopes present determines the immunogenicity of glutaraldehyde-inactivated PTx. Moreover, commercial batches of PTx and PTd intermediates exhibited minor variations in neutralizing epitopes. Conclusions: These findings have significant implications for developing acellular pertussis vaccines as they highlight the importance of preserving the neutralizing epitopes of PTx during detoxification to ensure the vaccine’s effectiveness. This assay is also valuable for the quality control of PTd as it more accurately represents the actual antigenic changes of PTx.

## 1. Introduction

Chemically and genetically inactivated pertussis toxin (PTx) is one of the most important antigenic components that compromise current acellular pertussis vaccines [1,2,3]. Glutaraldehyde, a protein cross-linking agent, has been widely used by many vaccine manufacturers to deactivate PTx over the past few decades. Glutaraldehyde could react actively with N-terminal amino groups of peptides, α-amino groups of amino acids, and sulfhydryl groups of cysteine, resulting in modifications of the surface of PTx and the formation of complex, high-molecular-weight protein species [4,5]. Our previous unpublished and others’ work has demonstrated that the structural changes impaired biochemical activities (residual toxicity) dependent on modifications of its two distinct functional domains: an A protomer (consisting of an S1 subunit) and a B oligomer (made up of S2, S3, S4, and S5 subunits) [6,7,8]. In addition, glutaraldehyde treatment also destroys the immunogenicity of PTx [9,10]. Detoxified PTx must maintain reasonable immunogenicity to ensure the high quality of acellular pertussis vaccines; thus, quality control of PT toxoid (PTd) is important.

It was proved that the overall structure changes could affect both linear and conformational epitopes, which can, in turn, influence immunogenicity [4,10]. Additionally, neutralizing antibodies directed against wild-type PTx fail to recognize chemically detoxified PTx [11]. Taken together, the integrity of the natural structure of PTx is critical for its antigenicity or immunogenicity. However, it is still unclear how the alterations in the structure of PTx could affect its antigenic properties and immunogenicity. Furthermore, how much the changes in antigenic characteristics could impair immunogenicity remained explored. In this context, an assay able to quantify antigenic properties of PTx could specifically indicate its quality post-detoxification.

Our unpublished work indicated that glutaraldehyde treatment causes dramatic structural changes in the A protomer and B oligomer, especially the S2 subunit. We hypothesize that the structural alterations impair neutralizing epitopes on the two functional domains. Accordingly, we established a sandwich ELISA analyzing antigenic properties of PTd preparations by using two monoclonal antibodies (mAbs), 1B7 against S1 and 11E6 against the S2/3 subunit as the detecting antibody. The two mAbs directly neutralize epitopes in the S1 and S2/3 regions of PTx, and their protective activities are demonstrated in vitro and vivo [12,13,14,15,16]. Subsequently, we applied the established ELISA assay to evaluate the retention of neutralizing epitopes on PTx with different degrees of detoxification. Then, the immunogenicity of acellular pertussis vaccine candidates containing various PTd preparations was evaluated by measuring serum anti-PTx-specific IgG. A correlation analysis of anti-PTx-specific antibody response and the number of neutralizing epitopes was performed. Finally, the assay was applied to assess the consistency of commercial batches of PTx and PTd intermediates. We are excited to share these notable findings in this manuscript and hope our work could give insightful indications for others working on developing and manufacturing acellular pertussis vaccines.

## 2. Materials and Methods

### 2.1. Preparation of PTd and PTd Containing Vaccine Formulation

To investigate the detoxification effect of glutaraldehyde concentration on PTx, various PTd preparations were produced as described but with modification [17]. In brief, purified PTx was detoxified with 0.5% glutaraldehyde in 50% glycerol for different times ranging from 10 min to 24 h. The detoxification process was performed at room temperature and terminated by adding sodium L-aspartate to a final concentration of 0.25 M. Subsequently, the detoxification mixture was diafiltered through a 30 kD membrane and then sterile filtered. To obtain acellular pertussis vaccine candidates, 25 μg PTd was absorbed into 1.3 mg/mL aluminum hydroxide.

### 2.2. Assay of Neutralizing Epitopes in S1 and S2/3 Subunits

Neutralizing epitopes in the S1 and S2/3 subunits were quantified using a sandwich ELISA established in this study. Rabbit polyclonal antibodies (pAbs) against PTd were prepared in the lab as previously described and used as the capture antibody [18]. Briefly, big-ear white rabbits, approximately 4 months of age and weighing about 3 kg, were injected with 0.6 mL of Freund’s complete adjuvant for sensitization. Two weeks later, 500 μg/mL PTx antigen was added to Freund’s complete adjuvant, mixed in a 1:1 volume, and fully emulsified. Then, 0.4 mL was injected into the thigh muscle, for a total of 0.8 mL. Three weeks later, 1 mL of the same antigen and adjuvant mix was injected subcutaneously on the back. After that, multiple injections were administered subcutaneously to the back, once every two weeks, to enhance immunization, a total of three times. Blood was collected 14 days after the third boost immunization, and immune sera were prepared by centrifugation.

Monoclonal antibodies (mAbs) specific to the S1 (1B7, NIBSC code 99/506) and S2/3 (11E6, NIBSC code 99/526) subunits served as the detecting antibody. In brief, 96-well microtitration plates (Greiner Bio-One, Fricken-hausen, Germany) were pre-coated with rabbit pAbs and incubated overnight at 4 °C. The plates were washed four times with PBS containing 0.05% Tween 20 as the washing buffer. For blocking, 100 μL of PBS containing 2% skim milk (assay buffer) was added to each well and incubated at 37 °C for 1 h. After aspirating the assay buffer, in-house reference (PTx) and PTd samples were prepared with the assay buffer and added to the wells. The starting concentration of PTx was set at 200 ng/mL to construct standard curves, with up to six serial dilutions used. Detecting mAbs were added according to the manufacturer’s instructions, followed by secondary peroxidase-labeled anti-mouse antibodies (Sera Care, Milford, MA, USA). The absorbance of each well was measured using a Multiskan FC reader (Thermo Fisher Scientific, Waltham, MA, USA) at 450/630 nm. The content of neutralizing epitopes in the S1 and S2/3 subunits was expressed in ng/mL. Subsequently, the assay’s specificity, accuracy, and precision were evaluated.

### 2.3. Immunization

Five- to six-week-old female mice (strain NIH) were injected intraperitoneally with a 0.5 mL one-third of a human single dose of vaccine candidate or PBS. Four weeks post-immunization, mice were anesthetized and bled for serum, and the serum was separated and stored at −20 °C until use. In all anti-PTx serological assays, the serum raised against PBS was the negative control, and the first international reference preparation (IRP) of mouse anti-PTx serum (97/642, NIBSC) was used as the positive control.

### 2.4. Assay of Measuring PTx-Specific Immunoglobulins (Ig), Total IgG

Antibodies against PTx were determined by ELISA assays as previously described [18,19,20]. Briefly, 96-well microtitration plates (Greiner Bio-One, Fricken-hausen, Germany) were pre-added with the purified PTx and kept overnight at 4 °C. The plates were washed 5 times with PBS containing 0.05% Tween 20. For blocking, 100 μL of PBS containing 1% bovine serum albumin (assay buffer) was added and incubated at 37 °C for one hour. After the assay buffer was aspirated, NIBSC reference serum (eight steps of 2-fold dilutions (1/100 to 1/409,600) were prepared with assay buffer) and Immune sera were added (diluted 2-fold for seven dilutions starting at 1/100), followed by the addition of secondary peroxidase-labeled anti-mouse antibodies (Sera Care, Milford, MA, USA). The absorbance of each well was read using a Multiskan FC reader (Thermo Fisher Scientific, Waltham, MA, USA) at 450/630 nm. The antibodies against PTx were expressed in IU/mL.

### 2.5. Ethics Statement for Animal Experiments

All animals used in this study were housed and cared for in facilities accredited by the Association for the Assessment and Accreditation of Laboratory Animal Care (AAALAC). The Institutional Animal Care and Use Committee (IACUC) approved all experiments.

## 3. Results

### 3.1. Establishment of the Neutralizing Epitopes Qualification ELISA Assay

#### 3.1.1. Working Range and Detection Limit

The 96-well plates were prepared as described in Section 2.2. To determine the working range and construct the standard binding curves, the initial concentration of PTx reference was set at 400 ng/mL, followed by eight serial 2-fold dilutions: 200, 100, 50, 25, 12.5, 6.25, and 3.125 ng/mL. In addition, the working concentrations of pAbs for coating were investigated. The working concentrations of mAbs against S1 and S2/3 were chosen according to the manufacturer’s recommendations. A linear regression analysis was performed by plotting the logarithm of the PTx reference concentration against the corresponding optical density (OD) reading. It was observed that the curve fitting was optimal within the concentration range of 6.25 ng/mL to 200 ng/mL, with the R^2^ values above 0.98 (Figure 1). In addition, OD readings of the PTx reference at 6.25 ng/mL were significantly above the cutoff values (Table 1). Therefore, five replicate experiments subsequently confirmed this working range, consistently yielding R^2^ values of 0.98 or higher (Appendix A). Thus, the working range was set between 6.25 and 200 ng/mL, and the lowest detection limit was determined as 6.25 ng/mL.

#### 3.1.2. Specificity

To validate the assay’s specificity in measuring epitopes in the S1 subunit, a batch of PTx was used as the positive control. The S2/4 subunit served as an interference control, while the assay buffer acted as the negative control. Similarly, to assess the assay’s specificity for the S2 subunit, the same batch of PTx and the S2/4 subunit were employed as the positive control, with the assay buffer as the negative control. The results showed that the OD values of the interference controls at high concentration and negative control were below the cutoff value; OD readings of the positive controls at low concentration were significantly above the cutoff value (Table 1), indicating reasonable specificity of the assay.

#### 3.1.3. Accuracy

To study the assay’s accuracy, spiked PTx samples were produced with concentrations of 100 (high value), 50 (middle high value), 25 (middle low value), and 12.5 ng/mL (low value) by diluting the PTx reference with the assay buffer. The measured values of the spiked samples were determined by testing three times. The recovery rate and relative bias were then calculated based on the measured and theoretical values. The results indicated that the three spiked samples had recovery rates between 70% and 130%, with a relative bias of less than 30% (Figure 1C,D), demonstrating good but not perfect accuracy.

#### 3.1.4. Precision

To assess the assay’s precision, we first evaluated its reproducibility by having the first experimenter analyze a batch of PTd samples six consecutive times. To evaluate the intermediate precision, the same PTd sample was tested by the second experimenter for another six consecutive times. The coefficient of variation (CV, %) of 12 results was calculated. The results showed that CVs were less than 30% (Figure 1E,F), indicating good but not perfect assay precision.

### 3.2. The Effect Glutaraldehyde Treatment on Antigenic Properties of PTx

The assay was conducted to analyze the changes in the antigenic properties of PTx following glutaraldehyde detoxification. Notably, as the degree of detoxification increased (with longer incubation time), there was no significant decreasing trend in the quantity of neutralizing epitopes in PTd (Figure 2A, Appendix A). However, the retention percentage of neutralizing epitopes in the S2/3 subunit showed a more significant decline, dropping from 47.0% to 16.7%, compared with the S1 subunit, where retention decreased from 65.6% to 25.7% (Figure 2B). This indicates that the S2/3 subunit is more sensitive to the degree of detoxification.

### 3.3. Correlation Between Neutralizing Epitope Content and Immunogenicity In Vivo

To investigate how much the retention percentage of neutralizing epitopes in PTx affects its immunogenicity and antigenicity, PTd preparations of different degrees of detoxification were absorbed into aluminum to produce acellular pertussis vaccine candidates. NIH female mice were immunized intraperitoneally with 1/3 human single dose (16.67 μg of PTd antigen) of vaccine candidates. The anti-PTx antibody level in immunized sera was analyzed as described in “Section 2 Materials and Methods”. As the concentration of glutaraldehyde increased, there was a decreasing trend in anti-PT IgG response (Figure 3A,B), indicating that the detoxification degree affects the immunogenicity of PTx. In addition, we observed a strong correlation between the anti-PTx IgG response and the neutralizing epitope in the S1 subunit (Pearson’s r = 0.59, Figure 3C), while the correlation with the neutralizing epitope in the S2/3 subunit was weak (Pearson’s r = 0.27, Figure 3D).

### 3.4. Monitoring Consistency of Neutralizing Epitopes in PTd Preparations

Our evaluation of the antigen quality and batch consistency of the PTx antigen component, based on the analysis of eight batches of purified PTx and PTd preparations produced by identical processes, has uncovered findings with significant potential implications. The retention percentage of neutralizing epitopes was a key focus. The number of neutralizing epitopes in S1 and S2/3 subunits of the purified PTx bulks ranged from 80% to 120% related to the in-house PTx reference (Figure 4A). In comparison, the epitopes in the S1 subunit varied between 24.5% and 40.7%, and in S2/3, the varied between 17.7% and 25.7% in PTd preparations (Figure 4B). These findings indicate that the assay is suitable for evaluating batch-to-batch consistency by quantifying variations in neutralizing antibody retention.

## 4. Discussion

Chemical or genetically detoxified PTx is a critical component of acellular pertussis vaccines, and the quality of PTx is essential for the vaccine’s efficacy. Chemical detoxification with glutaraldehyde causes significant structural changes in PTx, leading to a reduction in bioactivities associated with the A protomer and B oligomer [6,7,8]. These structural alterations can also affect the antigen properties of PTx, potentially compromising its immunogenicity [9,10]. Therefore, evaluating the antigenic properties of PTx is an important topic for quality control of an acellular pertussis vaccine.

In this manuscript, we introduced a sandwich ELISA assay by using two mAbs, 1B7 against S1 and 11E6 against S2/3 subunits, which accurately quantifies the number of neutralizing epitopes in the S1 and S2/3 subunits and provides a precise reflection of the antigenic changes in glutaraldehyde-inactivated PTx. This study started with the establishment and validation of the ELISA assay. Through multiple experiments, we determined the linear working range and detection limits. The assay demonstrated a wide working range from 6.25 ng/mL to 200 ng/mL, with a reasonable sensitivity as low as 10 ng/mL. Following that, the assay showed exciting specificity to the S1 subunit without cross-reactivity with subunits of B oligomer. However, we did not demonstrate the cross-reactivity of 11E6 with S1, because we failed to separate S1 from S2 during purification. Notably, Sato, H., and colleagues have already proven the excellent specificity of 1B7 and 11E6 [16]. The assay also exhibits good accuracy, with a recovery rate of spiked samples ranging from 70% to 130%. Additionally, the coefficient of variation of each measured value relative to the theoretical value was less than 30%, demonstrating good precision.

After its establishment, the assay was first used to assess the quality of PTd preparations with varying levels of detoxification. The assay was sensitive to detect the difference in the number (or retention percentage) of neutralizing epitopes in these PTd preparations that share the same protein content. Moreover, the immunogenicity of PTd preparations was positively correlated with the number of neutralizing epitopes retained. However, it is striking that weak correlations were found between the anti-PTx IgG response and the neutralizing epitopes in the S2/3 subunit compared with the strong correlations observed for the neutralizing epitopes in the S1 subunit. This could be explained by the fact that more neutralizing epitopes were preserved in the S1 subunit of PTx post-detoxification by glutaraldehyde; in other words, the anti-PTx-specific antibody responses were triggered mainly by neutralizing epitopes retained in the S1 subunit of PTd [9].

Later, the assay was used to evaluate the quality of the purified PTx bulks and PTd intermediates. Notably, these purified PTx bulks showed minor variations in the number of neutralizing epitopes, indicating good consistency between batches. Furthermore, different PTd intermediates also displayed antigenic properties with slight variations; specifically, the retention percentages of neutralizing epitopes in S1 and S2/3 subunits of PTd preparations were around 30% and 20% relative to PTx, respectively, demonstrating a stable chemical detoxification process by glutaraldehyde.

Assays for the final lot of the acellular pertussis vaccine can be conducted using in vivo methods to evaluate serology in mice or guinea pigs. However, these methods do not apply to PTd preparations due to time constraints [19]. Specifically, the antigenicity and immunogenicity of PTd should be evaluated in animal models as a vaccine formulation by measuring the level of anti-PTx-specific IgG in serum from immunized mice or guinea pigs; the process involves animal immunization, serum collection, and IgG detecting, which are time-consuming and not suitable for monitoring the quality and release of PTd in a timely and efficient manner. However, protein content quantification is commonly performed for assaying antigen components of acellular pertussis vaccines marketed in mainland China, which is essential but insufficient for identifying the masked epitopes caused by glutaraldehyde treatment. In this context, we and others have proposed a single radial diffusion (SRD) technique for measuring the total antigen content of PTx, FHA, and PRN bypasses aldehyde-induced masking of epitopes [18,21]. Although the SRD technique is valuable for batch consistency monitoring, it cannot distinguish between intact and structurally compromised antigens. Unlike the SRD assay, the ELISA assay in this manuscript specifically targets functional neutralizing epitopes critical for protective immunity. It offers a more refined assessment of antigenic quality, linking detoxification-induced structural changes to immunogenicity.

Additionally, many vaccine manufacturers prefer to use an ELISA, which is more efficient in quantifying PTx and evaluating the consistency of PTd preparations [22,23,24]. The integrity of the natural structure of PTx is greatly destroyed by glutaraldehyde treatment, resulting in a significant loss of some critical neutralizing epitopes in S1, S2, and/or S3, contributing to the protection in vivo [12,25,26]. For assay purposes, it is more logical to analyze changes in neutralizing epitopes of PTx after detoxification as this more accurately represents the actual antigenic changes of PTx.

According to our current understanding and the limited data available, we must acknowledge that the accuracy and precision of the assay are not perfect. Three factors may impact the reliability of the assay. Point 1: The concentration of the sample during detection is crucial. Specifically, when the sample is diluted to a concentration between 25 ng/mL and 50 ng/mL, the coefficient of variation (CV) of the results is smaller. Point 2: There is a potential loss of concentration of the in-house standard reference during storage. This standard was divided into Eppendorf tubes at a high concentration of around 1 mg/mL and stored at −20 °C for less than 3 months. We observed that prolonged storage time affected the quality of the PTx, including its concentration and the number of epitopes. Point 3: Significant dilution factors can lead to inaccurate dilutions. Both the in-house standard reference and the samples are initially prepared at a high concentration of approximately 1 mg/mL. These should be diluted to a range of 12.5 ng/mL to 100 ng/mL for detection, necessitating dilution factors between 100,000 and 800,000. To improve the assay, we are considering the following strategies: (1) Dilute the samples to a concentration between 25 ng/mL and 50 ng/mL each time. (2) Prepare a lyophilized standard reference (supplemented with a protein stabilizer) at a lower concentration, around 1 μg/mL, and store it in glass vials.

In addition, during this study, we observed that an increase in the concentration of glutaraldehyde was accompanied by a decrease in the antigenic properties and immunogenicity of the antigen. Notably, in our unpublished work, we demonstrated that adjustments to the pH significantly influence the ADP-ribosyltransferase activity of the A protomer. In contrast, variations in glutaraldehyde concentration have a greater impact on the B oligomer. Impairment of antigenic properties was accompanied by changes in biochemical activities (toxicity of PTx), highlighting that adjusting the pH and concentration of glutaraldehyde in the inactivation process could achieve a balance between the efficacy and safety of PTd.

The resurgence of pertussis in recent years has prompted the need for more effective pertussis vaccines, primarily because the effectiveness of acellular pertussis vaccines (which contain PTd) tends to wane over time [1,27,28,29,30,31]. Numerous previous preclinical and clinical studies have demonstrated that the safety, immunogenicity, and effectiveness of genetically detoxified pertussis toxin (gPT, with two amino acid substitutions, Arg9/Lys and Glu129/Gly) are significantly superior to those of chemically detoxified pertussis toxin (PTd) [32,33,34,35,36,37,38,39,40]. In this context, acellular pertussis vaccine containing gPT is a good choice for the next generation of pertussis vaccines. Compared with PTd, gPT displays a nearly identical conformational structure to native PTx [41]; it can skew cellular-mediated responses toward a Th1/Th17 phenotype [42,43,44], whereas a vaccine containing PTd is prone to inducing a Th2 lineage response [44,45]. gPT also triggers a long-lived humoral immune response, inducing longer-lasting protection, and its intrinsic adjuvant capacity enhances the immune response to other antigens, resulting in a strong immune response [27,41,46]. Notably, gPT is only successfully purified from the culture supernatant of a gene-modified Bordetella pertussis; currently, there is no recombinant expression of gPT available. In that case, manufacturing gPT on a large scale should consider the cost. Additionally, the genome stability of genetically modified seeds carrying two amino acid substitutions should be rigorously studied, verified, and monitored in case of a revert mutation back to the wild type.

## 5. Conclusions

To conclude, we established an in vitro assay that precisely reflects the antigenic changes of PTx post-glutaraldehyde detoxification. The antigenic properties of PTd preparations are closely related to immunogenicity. We also demonstrated that the assay is a potential alternative for monitoring the quality of PTx and PTd preparations in manufacturing high-quality acellular pertussis vaccines.

## Figures and Tables

**Figure 1 vaccines-13-00817-f001:**
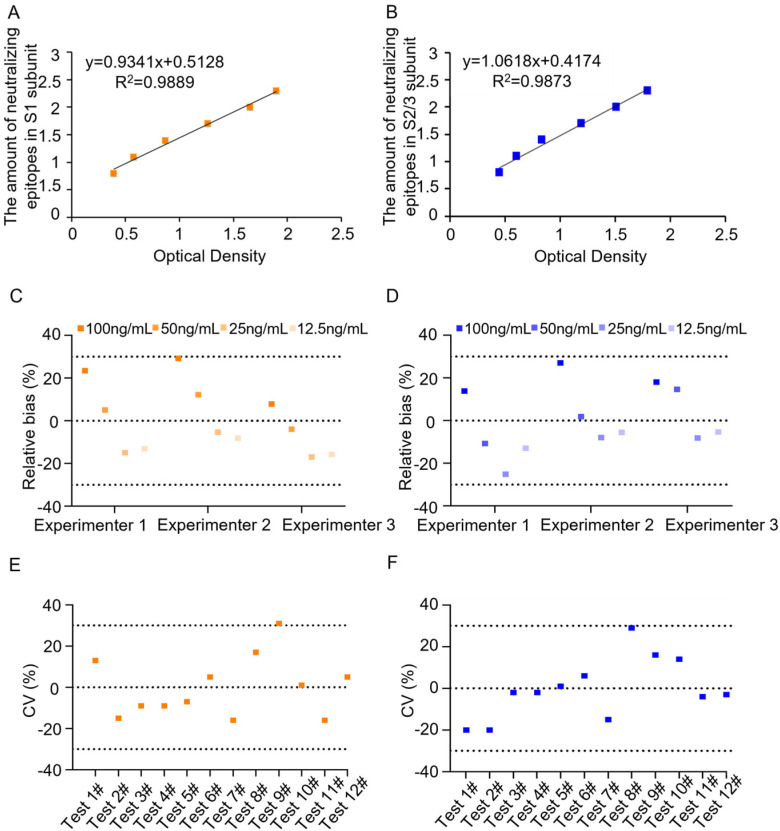
Establishment and primary validation of the neutralizing epitope qualification ELISA assay. To determine the working range and detection limit, both in-house reference and test samples were loaded with concentrations ranging from 6.25 to 200 ng/mL, and dose–response curves were plotted on the number of neutralizing epitopes (Log10) in S1 subunit (**A**) or S2/3 subunit (**B**) against the OD value. Following this, the precision of the assay for qualifying neutralizing epitopes in S1 and S2/3 was investigated (**C**,**D**). Meanwhile, the accuracy of the assay was also evaluated by analyzing PTx samples of low, middle, and high concentration (**E**,**F**). The recovery rate was expressed as relative bias.

**Figure 2 vaccines-13-00817-f002:**
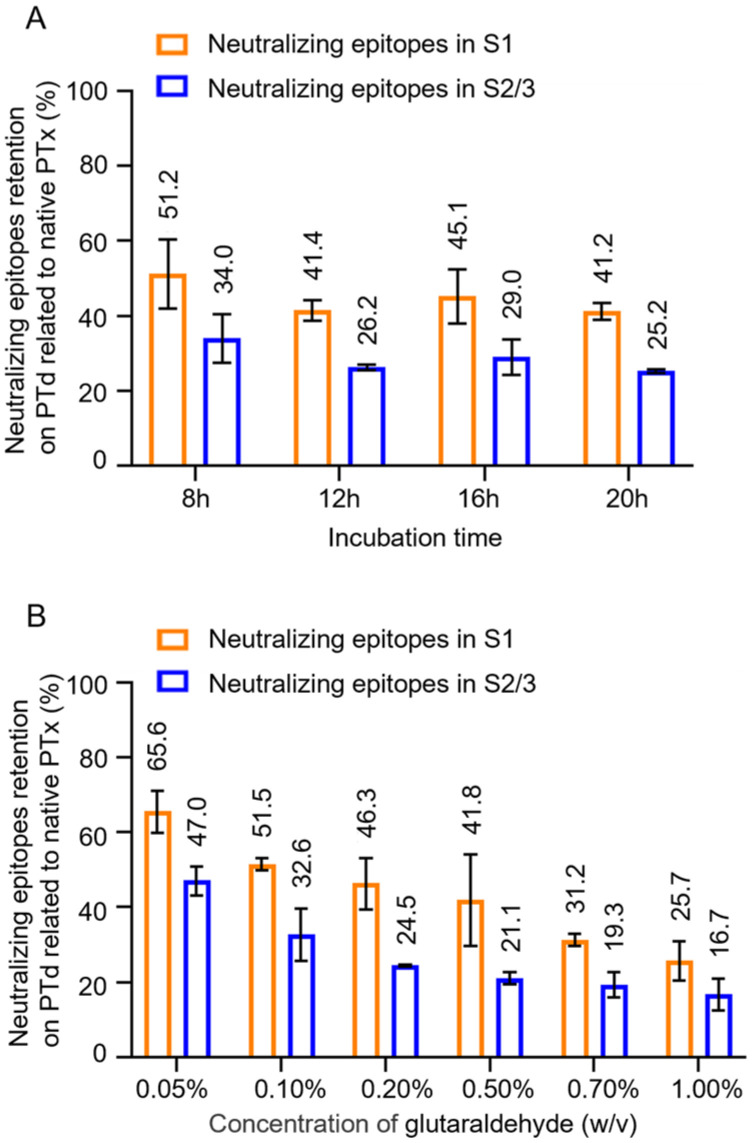
Effect of glutaraldehyde inactivation on neutralizing epitopes in the S1 and S2/3 subunits. To study the effect of incubation time on PTx, which had been inactivated by 0.5% (*w*/*v*) glutaraldehyde at room temperature (RT) for different times ranging from 8 to 20 h, to investigate the effect of concentration of glutaraldehyde, the purified PTx was treated with varying concentrations of glutaraldehyde ranging from 0.05% to 1.0% (*w*/*v*) at RT for 4 h. Subsequently, neutralizing epitope retentions in the S1 and S2/3 subunits of these PTd preparations were determined accordingly (**A**,**B**). The retention of neutralizing epitopes on PTd was presented as the percentage (%) related to native PTx.

**Figure 3 vaccines-13-00817-f003:**
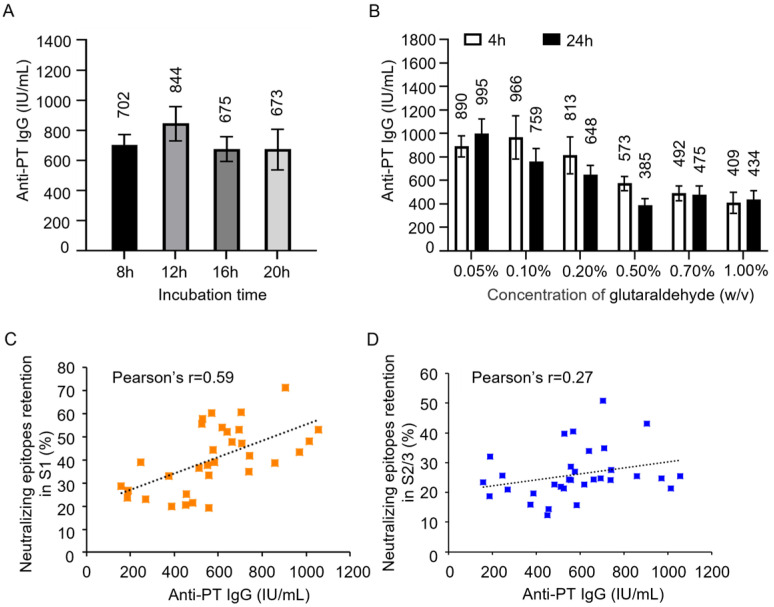
Effect of glutaraldehyde treatment on immunogenicity of PTx and correlation analysis. Various PTd samples were absorbed into aluminum to formulate acellular pertussis vaccine candidates. Immune serum was obtained from mice immunized with these different vaccine candidates. Following this, the levels of anti-PT-specific IgG were evaluated (**A**,**B**). A correlation analysis was performed between the level of anti-PT IgG and the number of epitopes in the S1 subunit (**C**) and the S2/3 subunit (**D**).

**Figure 4 vaccines-13-00817-f004:**
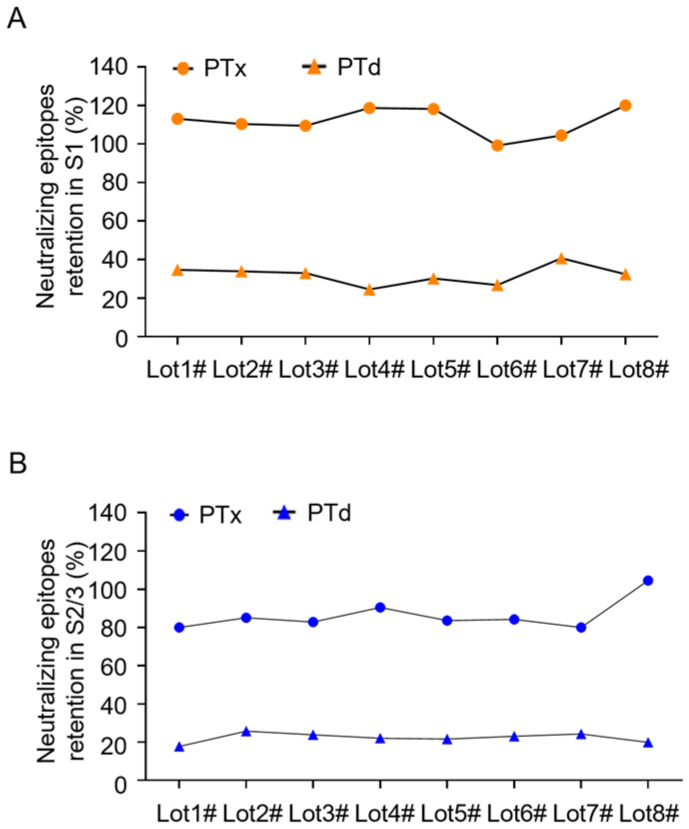
Monitoring the number of neutralizing epitopes in consistent batches of PTx bulks and PTd intermediates. To investigate its application for in-process control, the assay was used to evaluate the number of neutralizing epitopes in the S1 subunit (**A**) and S2/3 subunit (**B**) of eight commercial, consistent batches of PTx bulks and PTd intermediates, respectively.

**Table 1 vaccines-13-00817-t001:** Validation on specificity of the assay.

Detecting Abs	Samples	Concentration (ng/mL)	OD Value	Cut-Off Value ^1^
1B7	PTx	6.25	0.372	0.347
	S24	5000	0.193	
	Assay buffer	N/A	0.165	N/A
11E6	PTx	6.25	0.353	0.306
	S24	6.25	0.358	
	Assay buffer	N/A	0.146	N/A

Note: ^1^ Cut-off value was calculated as 2.1 times the OD value of assay buffer.

## Data Availability

Additional data will be made available upon request.

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
