# Peer review of "The Immunogenicity of Glutaraldehyde Inactivated PTx Is Determined by the Quantity of Neutralizing Epitopes"

_vaccines, 2025, doi:10.3390/vaccines13080817_

Round 1

Reviewer 1 Report

Comments and Suggestions for Authors

Please refer to the file attached

Comments on the Quality of English Language

Acceptable

Author Response

Dear reviewer,

We sincerely appreciate your time to review our manuscript and provide valuable feedback. Your insightful comments have led to meaningful improvements in this version. The authors have carefully considered each comment and made every effort to address them. We hope that, after these revisions, the manuscript meets your high standards. We welcome any further constructive feedback you may have. Below, we present our point-by-point responses. All modifications in the manuscript have been highlighted in yellow.

Comment 1: P8. Explain more specifically “---do not apply to PTd preparation due to time constraints” What you mean by time constraints?

Response 1: Thank you for pointing this out; I appreciate your valuable comment. We want to clarify a few points here. Currently, in mainland China, to evaluate the immunity and antigen quality of PTd, it should be formulated as an acellular pertussis vaccine and tested in animals. This approach is time-consuming and complicated (first, immunize the animals with vaccine candidates; then collect serum, and finally, measure anti-PTx specific IgG by ELISA). It is not suitable for monitoring the quality and release of PTd in a timely and efficient manner. We have added the explanation in the discussion part of the revised manuscript- Page 9, Lines 285-290 [ Specifically, the antigenicity and immunogenicity of PTd should be evaluated in animal models as a vaccine formulation by measuring the level of anti-PTx-specific IgG in serum from immunized mice or guinea pigs, the process involved animal immunization, serum collection and IgG detecting, which is time-consuming and not suitable for monitoring the quality and release of PTd in a timely and efficient manner.]

Comment 2: Fig. 3 and 4. Glutaraldehyde treatment inevitably lowers the immunogenicity of S1/S2/3 antigens, as expected. Please discuss how to strike the balance between efficacy and safety of PTd by inactivation process.

Response 2: This is a good point, and you are right. Many factors could impact the quality of PTx during glutaraldehyde inactivation. This manuscript/work focuses on evaluating the antigenicity of PTd and the relationship between neutralizing epitopes and antigenicity. Therefore, we did not discuss the balance between efficacy and safety of PTd through the inactivation process in detail. In the other manuscript (under review) of our team titled "Different modification arts of glutaraldehyde on two functional domains of pertussis toxin, impacting its toxicity and immunogenicity," we mainly investigated the effect of pH and concentrations of glutaraldehyde on the toxicity (safety) and antigenicity (efficacy) of PTx. We found that adjustments to the pH significantly influence the ADP-ribosyltransferase activity of the A protomer. At the same time, variations in glutaraldehyde concentration have a greater impact on the B oligomer. Impairment of antigenic properties was accompanied by changes in biochemical activities (toxicity of PTx), highlighting that adjusting the pH and concentration of glutaraldehyde in the inactivation process could achieve a balance between the efficacy and safety of PTd. In the discussion section of the revised version of the manuscript, we added a paragraph discussing a little bit on how to strike the balance between efficacy and safety of PTd by the inactivation process- Pages 9-10, Lines 325-333 [ In addition, during this study, we observed that an increase in the concentration of glutaraldehyde was accompanied by a decrease in the antigenic properties and immunogenicity of the antigen. Notably, in our unpublished work, we demonstrated that adjustments to the pH significantly influence the ADP-ribosyltransferase activity of the A protomer. In contrast, variations in glutaraldehyde concentration have a greater impact on the B oligomer. Impairment of antigenic properties was accompanied by changes in biochemical activities (toxicity of PTx), highlighting that adjusting the pH and concentration of glutaraldehyde in the inactivation process could achieve a balance between the efficacy and safety of PTd.].

Comment 3: Discussion. The chemical inactivation significantly hampers the protective epitopes, and it is well noted that the duration of the immunogenicity is short, even after multiple (5~6 time) immunization during childhood. In this respect, it is worth discussing on extending the present study to genetically detoxified PTd, and the pros and cons between genetic vs chemical detoxification with a view to further improve the pertussis vaccine.

Response 3: That is a valid and interesting point. As you know, the resurgence of pertussis in recent years has prompted the need for more effective pertussis vaccines, primarily because the effectiveness of acellular pertussis vaccines (which contain PTd) tends to wane over time. In this context, acellular pertussis vaccine containing gPT is a good choice for the next generation of pertussis vaccines. Of course, both gPT and PTd have pros and cons; in the revised manuscript, we added some discussion on this topic- Page 10, lines 334-351 [ The resurgence of pertussis in recent years has prompted the need for more effective pertussis vaccines, primarily because the effectiveness of acellular pertussis vaccines (which contain PTd) tends to wane over time. Numerous previous preclinical and clinical studies have demonstrated that the safety, immunogenicity, and effectiveness of genetically detoxified pertussis toxin (gPT, with two amino acid substitutions: Arg9/Lys and Glu129/Gly) are significantly superior to those of chemically detoxified pertussis toxin (PTd). In this context, acellular pertussis vaccine containing gPT is a good choice for the next generation of pertussis vaccines. Compared to PTd, gPT displays a nearly identical conformational structure to native PTx; it can skew cellular-mediated responses toward a Th1/Th17 phenotype, whereas a vaccine containing PTd is prone to inducing a Th2 lineage response. gPT also triggers a long-lived humoral immune response, inducing longer-lasting protection, and its intrinsic adjuvant capacity enhances the immune response to other antigens, resulting in a strong immune response. Notably, gPT is only successfully purified from the culture supernatant of a gene-modified Bordetella pertussis; currently, there is no recombinant expression of gPT available. In that case, manufacturing gPT on a large scale should consider the cost. Additionally, the genome stability of genetically modified seeds carrying two amino acid substitutions should be rigorously studied, verified, and monitored in case of a revert mutation back to the wild type.] In addition, new references were added in the revised manuscript- Page 10, lines 449-512

Reviewer 2 Report

Comments and Suggestions for Authors

Chemical detoxification using glutaraldehyde typically results in significant structural changes to the toxin. However, how these structural changes in PTx affect its antigenic properties remain unclear. To investigate the impact of structural changes on antigenic properties, the authors developed a sandwich ELISA to quantify the neutralizing epitopes on PTx. Using the ELISA-based assay, they analyzed different PTx toxoid (PTd) preparations, assessed the immunogenicity of various acellular pertussis vaccine candidates containing these PTd preparations, and evaluated the consistency of commercial batches of PTx and PTd intermediates. They found that the assay was sensitive enough to quantify variations in neutralizing epitopes among different PTd samples that shared the same protein concentration. Importantly, they found a positive correlation between the number of neutralizing epitopes in detoxified PTx and its immunogenicity, indicating that the amount of neutralizing epitopes present determines the immunogenicity of glutaraldehyde-inactivated PTx. Moreover, they found that commercial batches of PTx and PTd intermediates exhibited minor variations in neutralizing epitopes. These findings have significant implications for the development of acellular pertussis vaccines, as they highlight the importance of preserving the neutralizing epitopes of PTx during detoxification to ensure the vaccine's effectiveness. However, I have the following suggestions and comments on this manuscript.

  1.      Line 89: Rabbit polyclonal antibodies against PTd were used as capture antibodies. Please list the source of the polyclonal antibody, although a reference was cited.
  2.      Line 93: The microtiter plate was pre-coated with murine pAbs. Please be consistent. Murine or rabbit?
  3.      Line 95: 2% milk was added to each well. 2% milk or 2% skim milk? They are quite different. Please be specific so that others can reproduce what you published.
  4. Table 1. What is the CV for each experiment? <30%?
  5.      Table 1. How was the cut-off defined?
  6.      Line 179: CVs were less than 30%, indicating good assay precision. A CV higher than 15% is usually not acceptable for most assays used for diagnosis. CV higher than 20% is generally not sufficient for production. Do you know the source of high CVs?
  7.      Figure 3: It would be better to include error bars.

Author Response

Dear reviewer,

We sincerely appreciate your taking the time to review our manuscript and provide valuable feedback. Your insightful comments have led to meaningful improvements in this version. The authors have carefully considered each comment and made every effort to address them. We hope that, after these revisions, the manuscript meets your high standards. We welcome any further constructive feedback you may have. Below, we present our point-by-point responses. All modifications in the manuscript have been highlighted in yellow.

Comment 1:  Line 89: Rabbit polyclonal antibodies against PTd were used as capture antibodies. Please list the source of the polyclonal antibody, although a reference was cited.

Response 1: Thank you for pointing this out. We have added the source of the polyclonal antibody and provided a brief protocol for its preparation. - Pages 2-3, Lines 89-99 [Rabbit polyclonal antibodies (pAbs) against PTd were prepared in the lab as previously described…. Blood was collected 14 days after the third boost immunization, and immune sera were prepared by centrifugation.].

Comment 2:  Line 93: The microtiter plate was pre-coated with murine pAbs. Please be consistent. Murine or rabbit?

Response 2: We apologize for any inconsistencies that may have affected your reading experience. We have corrected this mistake and highlighted it in the revised manuscript-Page 3, Lines 102-103 [pre-coated with rabbit pAbs].

Comment 3:  Line 95: 2% milk was added to each well. 2% milk or 2% skim milk? They are quite different. Please be specific so that others can reproduce what you published.

Response 3: Yes, we agree with you; this is a typographical error. We used 2% skim milk, not milk. We have corrected it in the revised manuscript -Page 3, Line 105 [ 2% skim milk (assay buffer) was added to each well].

Comment 4: Table 1. What is the CV for each experiment? <30%?

Response 4: - Thank you for your comment. Here is the original data, as shown in the table below, where all the CVs are less than 20%.

Table1. Validation of the specificity of the assay

Detecting Abs

Samples

Concentration (ng/mL)

Repeat 1

Repeat 2

Repeat 3

calculated OD value

CV%

Cut-off value

1B7

PTx

6.25

0.386

0.423

0.308

0.372

15.8%

0.347

S24

5000

0.191

0.195

/

0.193

1.5%

Assay buffer

N/A

0.189

0.142

/

0.165

19.9%

N/A

11E6

PTx

6.25

0.375

0.339

0.346

0.353

5.4%

0.306

S24

6.25

0.361

0.354

0.360

0.358

1.1%

Assay buffer

N/A

0.147

0.144

/

0.146

1.5%

N/A

Comment 5:   Table 1. How was the cut-off defined?

Response 5: We apologize for the missing information which we should have specified in the manuscript. The cut-off value was defined as 2.1 times the OD value of the assay buffer; we have added this note information under Table 1 in the revised manuscript-Page 4, Line 157 [ Note: 1. Cut-off value was calculated as 2.1 times the OD value of assay buffer.].

Comment 6: Line 179: CVs were less than 30%, indicating good assay precision. A CV higher than 15% is usually not acceptable for most assays used for diagnosis. CV higher than 20% is generally not sufficient for production. Do you know the source of high CVs?

Response 6: Yes, we agree with you. The smaller the CVs, the more reliable the assay is. In this context, the accuracy and precision of the assay were good but not perfect, which should be improved in the future.  In addition, we added our discussions on the source of the assay and the strategies to improve the assay in the updated manuscript – Page 9, lines 308-324 [According to our current understanding and the limited data available, we must acknowledge that the accuracy and precision of the assay are not perfect. Three factors may impact the reliability of the assay. Point 1: The concentration of the sample during detection is crucial. Specifically, when the sample is diluted to a concentration between 25 ng/mL and 50 ng/mL, the coefficient of variation (CV) of the results is smaller. Point 2: There is a potential loss of concentration of the in-house standard reference during storage. This standard was divided into eppendorf tubes at a high concentration of around 1 mg/mL and stored at -20°C for less than 3 months. We have observed that prolonged storage time affects the quality of the PTx, including its concentration and the number of epitopes. Point 3: Significant dilution factors can lead to inaccurate dilutions. Both the in-house standard reference and the samples are initially prepared at a high concentration of approximately 1 mg/mL. These should be diluted to a range of 12.5 ng/mL to 100 ng/mL for detection, necessitating dilution factors between 100,000 and 800,000. To improve the assay, we are considering the following strategies: (1) Dilute the samples to a concentration between 25 ng/mL and 50 ng/mL each time; (2) Prepare a lyophilized standard reference (supplemented with a protein stabilizer) at a lower concentration, around 1 μg/mL, and store it in glass vials.].

Comment 7:  Figure 3: It would be better to include error bars.

Response 7: Thank you for your valuable comment. I would like to clarify some information with you. Figure 3 in the manuscript already includes error bars. I believe the figure you were referring to is actually Figure 4. I apologize for any confusion. We are unable to generate error bars for Figure 4 because each data point represents only one batch or lot of PTx or PTd.

Round 2

Reviewer 2 Report

Comments and Suggestions for Authors

The authors answered all my critiques, and the revised manuscript is significantly improved.